# Associations between the *VDR* Gene rs731236 (TaqI) Polymorphism and Bone Mineral Density in Postmenopausal Women from the RAC-OST-POL

**DOI:** 10.3390/biomedicines12040917

**Published:** 2024-04-20

**Authors:** Sylwia Górczyńska-Kosiorz, Elżbieta Tabor, Paweł Niemiec, Wojciech Pluskiewicz, Janusz Gumprecht

**Affiliations:** 1Department of Internal Medicine, Diabetology and Nephrology, Faculty of Medical Sciences in Zabrze, Medical University of Silesia, 40-055 Katowice, Poland; etabor@sum.edu.pl (E.T.); jgumprecht@sum.edu.pl (J.G.); 2Department of Biochemistry and Medical Genetics, School of Health Sciences in Katowice, Medical University of Silesia, Medyków Street 18, 40-752 Katowice, Poland; pniemiec@sum.edu.pl; 3Metabolic Bone Diseases Unit, Department of Internal Medicine, Diabetology and Nephrology, Faculty of Medical Sciences in Zabrze, Medical University of Silesia, 40-055 Katowice, Poland; osteolesna@poczta.onet.pl

**Keywords:** bone mineral density, gene polymorphism, vitamin D receptor, osteoporosis, postmenopausal

## Abstract

Background: Postmenopausal osteoporosis is not only related to hormonal factors but is also associated with environmental and genetic factors. One of the latter is the polymorphism of vitamin D receptor (*VDR*). The aim of the reported study was to comprehensively analyze the *VDR* gene polymorphic variants rs731236 (TaqI), rs1544410 (BsmI) and rs7975232 (ApaI) in the Polish population of postmenopausal women. Methods: The study group consisted of 611 women after menopause (their median age was 65.82 ± 6.29 years). Each of them underwent bone densitometry (DXA) of the non-dominant femoral neck and total hip with a biochemical analysis of vitamin D3 serum concentration and genotyping of the above-mentioned single nucleotide polymorphisms (SNPs); the obtained results were analyzed in the aspect of waist circumference (WC), body mass index (BMI) and past medical history. Results: The genotype prevalence rates of all SNPs were compatible with Hardy–Weinberg equilibrium (*p* > 0.050). Out of the studied polymorphisms, only rs731236 genotype variants affected DXA, with AG heterozygotes showing the worst bone parameters. Neither patient age nor vitamin D3 concentration, BMI, WC or comorbidities was associated with rs731236 genotype. Conclusions: Out of the polymorphisms studied, only rs731236 genotypes differed among the DXA results, while the AG heterozygotes were characterized by the lowest median bone mineral density.

## 1. Introduction

Hormonal changes that occur in postmenopausal women may lead to the development of various medical conditions. The extinction of ovarian reproductive function and estrogen deficiency are risk factors for metabolic diseases such as obesity, diabetes, and osteoporosis (OP), as well as cardiovascular disorders [1]. Regarding postmenopausal women, estrogen deficiency is a major, though not the only, cause of bone mass loss. Estrogens act as regulators of bone metabolism. Estrogen deficiency enhances the activity of osteoclasts, triggering the process of bone demineralization, which, with decreased calcium absorption in the intestines, has adverse effects on the maintenance of bone mass [2,3]. Among the factors that promote the development of OP are genetic, hormonal and environmental aspects, including physical inactivity, improper lifestyle, harmful habits (smoking, alcohol abuse), nutritional deficiencies (insufficient intake of vitamin D and calcium) and the intake of certain medications like heparin, methotrexate, antiepileptic drugs, GnRH (gonadotropin-releasing hormone) analogs, T3 and T4 preparations, vitamin K antagonists and tamoxifen [1,4,5].

Bone mineral density (BMD) remains an important diagnostic marker of OP and a strong risk factor for fractures [6,7]. It is estimated that up to 85% of the BMD-affecting factors are genetic ones [6]. Numerous BMD-associated loci have been identified in genome-wide association studies (GWASs), demonstrating both common and rare variants among the candidate genes associated with BMD [1,7], one of them being the vitamin D receptor gene (*VDR*), recognized as a potential factor contributing to OP [8,9]. VDR is an intracellular hormone receptor found in many tissues. Its regulation is influenced by genetic and environmental factors such as diet, sunlight and infections. As estrogen increases the *VDR* gene expression in bones, its menopause-related deficiency may lead to an opposite effect [10,11].

The mechanism of VDR activation involves the formation of a heterodimer with the retinoid X receptor (RXR) and binding to 1,25(OH)2D. The complex migrates to the nucleus, affecting transcriptional regulation. VDR influences the expression of 2–3 thousand genes, including those related to mineral and bone metabolism, such as the genes associated with the receptor activator of nuclear factor kappa-B (RANK receptor), receptor activator of nuclear factor kappa-B Ligand (RANKL), calcium transporters, osteocalcin, osteopontin and alkaline phosphatase, as well as genes connected to other metabolic pathways related to the pleiotropic effects of vitamin D [12,13,14].

The *VDR* gene is located on the long arm of chromosome 12 (locus 12q13.11). The gene is 75 kbp in size and includes 12 exons. More than 900 allelic variants have been reported in the *VDR* gene area, including those that cause vitamin D resistance or affect *VDR* gene expression levels, thus either increasing or decreasing the vitamin D effects on specific cells. The polymorphic variants rs731236 (TaqI), rs1544410 (BsmI) and rs7975232 (ApaI) are located in the 3’UTR region, which is responsible for the stability of the resulting mRNA [10] (Appendix A). It appears that genetic variation in the *VDR* gene may also affect pharmacogenetics [9,13]. VDR genetic variants are likely to modulate the effects of vitamin D supplementation [14]. An association with vitamin D levels has been demonstrated for several polymorphic variants of the *VDR* gene, including rs2228570 (known as FokI), rs731236 (known as TaqI) and rs11568820 [15,16]. A higher risk for OP has been shown with possession of the G allele for the rs731236 variant, especially in the homozygous form, as observed in a young Saudi female populations [17].

Metabolic syndrome (MetS) and OP are apparently unrelated, but abdominal obesity, hyperglycemia, dyslipidemia and hypertension are the risk factors for OP, being also the components of MetS. On the one hand, abdominal obesity is associated with osteoporosis, while overweight and BMI-related protective effects against bone loss are also described on the other [18]. Variants of the *VDR* gene have also been studied as risk factors and molecular markers of metabolic syndrome, but the results have not been consistent [19]; rs1544410 and rs7975232 were found to be risk factors for the development of type 1 diabetes [20], and a meta-analysis found rs1544410 to be protective in MetS [19].

Therefore, the aim of the reported study was to comprehensively analyze the polymorphic variants of the *VDR* gene, i.e., rs 731236 (TaqI), rs1544410 (BsmI) and rs7975232 (ApaI), in menopausal women by searching for genetic markers to identify patients at a higher risk of developing osteoporosis in the context of their comorbidities. The effects of *VDR* gene polymorphisms on the clinical phenotype of the women studied, including BMD, were analyzed in the context of comorbidities, medical history data and other clinical factors.

## 2. Materials and Methods

This retrospective cohort study was a part of an epidemiological project called the RAC-OST-POL study and was approved by the Bioethical Committee of the Medical University of Silesia (No. KNW/0022/KB1/9/I/10). The study was conducted in accordance with the STROBE guidelines. The baseline epidemiologic characteristics of the study group were presented earlier [21].

### 2.1. Patient Recruitment and Examination

The participants of the research project were randomly recruited from the general population of over 17,500 postmenopausal women aged over 55 years (mean menopausal age in the Polish population is 49 years) inhabiting Racibórz, a town in the Upper Silesia region of southern Poland. The invitations to the project were sent via regular post to 1750 subjects (10% of the population in the above-mentioned sex and age group). As many as 625 women answered to the invitation letters. All the participants gave informed consent to participate. Only the participants with complete results of all the genotyping stages were included in the final analysis. The final number of participants was 611 women.

The medical history of comorbidities, fractures, medications and cigarette and alcohol abuse were obtained and analyzed from all the study participants. Body weight and height were also measured in each participant, and BMI was calculated.

### 2.2. Bone Mineral Density Measurement

Bone density parameters were measured with the use of a Lunar DPX (GE Healthcare, Waukesha, WI, USA) device. Bone mineral densities (BMDs) of the non-dominant femoral neck (FN) and total hip (TH) were evaluated in each patient and presented in standardized units [g/cm^2^], based on corresponding T-score values according to the National Health and Nutrition Examination Survey (NHANES) reference data for white women (aged between 20 and 29 years). The WHO criteria were used for osteoporosis diagnosis. All the measurements were performed by one experienced operator. The coefficient of variation (CV%), calculated per 50 measurements, was 1.6% for FN and 0.82% for TH.

### 2.3. Biochemical and Genetic Analyses

Serum vitamin D3 concentrations were assayed. After two days of column extraction, a 25-OH-Vitamin D ELISA kit (Immundiagnostik, Bensheim, Germany) was used.

Venous blood samples were collected from each participant and frozen at a temperature of −20 °C until DNA isolation. Genomic DNA was isolated using a MasterPure genomic DNA purification kit (Epicenter Technologies, Madison, WI, USA). Three single nucleotide polymorphisms (SNPs) of the *VDR* gene, namely rs731236 (TaqI), rs7975232 (ApaI) and rs1544410 (BsmI), were genotyped using TaqMan Predesigned SNP Genotyping Assay kits and the 7300 Real-Time PCR System (Thermo Fisher Scientific, Foster City, CA, USA). The accuracy of genotyping was verified by re-genotyping 10–15% of samples. The repeatability of results reached 100%.

### 2.4. Statistical Analysis

Statistical analyses were performed using Statistica 13.1 (TIBCO Software Inc., Palo Alto, CA, USA). Data distribution was checked with the Shapiro–Wilk test. Since all quantitative data were non-normally distributed, the data were presented as medians and their spreads as quartile deviations (QDs). In the case of a dichotomous grouping of variables, the Mann–Whitney U test was used. When three or more groups were compared for a given quantitative variable, the Kruskal–Wallis test was used, along with a post-hoc analysis. The Spearman’s r_s_ coefficient was used to interpret the strength of the correlation between quantitative variables. Hardy–Weinberg equilibrium was tested by the χ^2^ test, as well as by the comparisons of genotype and allele prevalence rates between the groups differentiated by qualitative variables. Fisher’s correction was used for the subgroups with fewer than ten subjects.

The analysis of factors being quantitative variables, such as the results of the DXA test (BMD FN and BMD TH, along with their T-scores), vitamin D3 concentration, BMI or waist circumference, consisted of comparing the values of their medians in the additive model of inheritance (between individual genotypes) and the recessive/dominant model (between carriers of individual alleles). The analysis of qualitative variables, such as a history of fractures, diagnosis of osteoporosis according to WHO criteria (T-score ≤ −2.5), the use of anti-osteoporosis therapy, the diagnosis of visceral obesity, diabetes and others, consisted of determining whether there were statistically significant differences in the prevalence rates of genotypes between particular classes of these variables.

The cases with missing data were rejected from the respective comparisons. A *p* value less than 0.050 was assumed as statistically significant. In the case of multiple comparisons, the *p* values were adjusted using the Bonferroni correction.

## 3. Results

### 3.1. General and Clinical Characteristics of the Study Group

The median age (±QD) of the participants was 65.82 ± 6.29 years. All the patients declared being at least 12 months past their last menstruation, which was defined as confirmed menopause (the median age of the menopausal patients was 50.00 ± 3.00 years). The median BMI was 30.84 ± 3.96 kg/m^2^. Other demographic, clinical and biochemical data are presented in Table 1.

As many as 163 (26.67%) participants revealed a positive family history of fractures (in 22.42%, fracture occurred in the mother, while in 5.07%, fracture occurred in the father). In 170 (27.82%) participants, fractures were diagnosed over the age of 40 years. The most frequent location was the forearm (104 fractures in 91 women). In 92 (15.06%) patients, anti-osteoporotic therapy was initiated, and 79 (12.93%) and 89 (14.57%) participants had vitamin D and calcium supplementation prescribed, respectively (the therapy duration was over 12 months).

The femoral neck and total hip BMD scores and the FN and TH T-scores are presented in Table 1. According to the WHO and ISCD criteria, based on the FN T-scores, osteoporosis was diagnosed in 57 (9.45%) patients and osteopenia in 335 (55.56%) patients. Figure 1 presents the prevalence rate (%) of the individual ranges of FN and TH T-scores, based on the WHO definition of osteoporosis and osteopenia, in the study group.

### 3.2. VDR Gene Polymorphisms

The genotype and allele prevalence rates of the analyzed *VDR* gene polymorphisms are presented in Table 2. The genotype prevalence rates of all the SNPs were compatible with the Hardy–Weinberg equilibrium (*p* > 0.050).

### 3.3. VDR Gene Polymorphisms and Bone Mineral Density

This section presents the results of the analysis of the *VDR* gene polymorphisms in the context of bone mineral density parameters.

Out of the three polymorphisms studied, only rs731236 (TaqI) genotypes demonstrated differences in DXA scores (see Figure 2). The AG heterozygotes each time presented the lowest median bone mineral density and the lowest median value of T-scores. In almost all the cases, statistically significant differences were observed between the AG heterozygotes and both types of homozygotes (see Figure 2).

Also in the recessive/dominant model (the carrier state analysis), differences were shown in some BMD parameters among the variants of the rs731236 (TaqI) polymorphism. The carriers of the A allele were characterized by significantly lower BMD FN values and lower T-scores of the parameter vs. the GG homozygotes (see Table 3).

### 3.4. Correlates and Modifiers of Bone Mineral Density

This section presents the results from the analysis of potential quantitative correlates of bone mineral density parameters as well as qualitative BMD modifiers.

Table 4 shows the Spearman correlation coefficient (r_s_) values for the BMD parameters and other clinically relevant quantitative variables. Only statistically significant results (*p* < 0.050) are shown.

Negative correlations of moderate strength were found between all the DXA parameters and the age of the subjects. The concentration of vitamin D3 was weakly correlated with BMD values and slightly more strongly, though negatively, correlated with age. Both BMI and waist circumference were positively correlated with BMD parameters, weakly correlated with BMD FN and its T-score, and moderately correlated with BMD TH and its T-scores (see Table 4).

In order to verify whether the age, vitamin D3 concentration, BMI and waist circumference may interfere with the results obtained in Section 3.3, it was examined whether those parameters differentiated the patients with individual genetic variants of the rs731236 (TaqI) polymorphism. There were no statistically significant differences between the genotypes and the carrier state variants. Interestingly enough, the patients with the GG genotype, characterized by the highest median BMD FN and the highest median BMD FN T-scores in the reported study, were older (median ± QD: 67.23 ± 5.74 years) than both the AG heterozygotes (median ± QD: 66.97 ± 6.78 years) and the AA homozygotes (median ± QD: 64.12 ± 5.88 years), but those differences were not statistically significant. The GG homozygotes were also older than the carriers of the A allele (median ± QD: 65.64 ± 6.47 years), who were characterized by worse DXA parameters in the recessive/dominant model (Section 3.3). In neither case were the differences statistically significant (*p* = 0.292).

Bone mineral density was also compared within the qualitative variables that are potential modifiers of BMD (see Table 5). Only the median BMD scores (FN and TH) are shown because the trends for T-scores were the same.

Higher bone mineral density was observed in obese women and women with visceral obesity (statistically significant differences in all DXA parameters) and in people with either type of diabetes (statistically significant differences for BMD TH and BMD TH T-score).

The factors associated with lower bone mineral density included rheumatoid arthritis (the differences were statistically significant only for BMD TH and BMD TH T-scores), calcium and vitamin D3 supplementation, anti-osteoporotic therapy and a positive history of fractures after the age of 40. Regarding those parameters, statistically significant differences were demonstrated for both femoral neck and total hip scores together with their T-scores.

The reported study showed no statistically significant effects of smoking, glucocorticoid therapy or family history of fractures on BMD (*p* > 0.050).

In a subsequent step, differences were analyzed in the prevalence rates of the genetic variants of the rs731236 (TaqI) polymorphism in the context of either the presence or the absence of each of the BMD-modifying factors (see Table 6). The results of those analyses, carried out in the additive and recessive/dominant models, indicate no such differences (*p* > 0.050), which may suggest that the BMD-modifying factors did not affect the results obtained in Section 3.3.

## 4. Discussion

Our study showed that AG heterozygotes presented lower bone density than subjects with other genotypes of the rs731236 (TaqI) polymorphism, both at the FN and TH. They accounted for 45% of the group (*n* = 275). The highest bone density values were observed in the GG homozygotes. No statistically significant differences in bone parameters were observed for the other VDR polymorphisms studied, i.e., either for rs7975232 (ApaI) or rs1544410 (BsmI).

Our results do not correspond to Elzbieta Jakubowska-Pietkieiwcz et al.’s [22] results for the comparison of rs731236, rs7975232 and rs1544410 allele prevalence rates. The study was conducted on the Polish population as well, but the age range was 6–18 years. Since the patients studied were young, it can be speculated that bone health was more influenced by genetic factors than by environmental aspects. Perhaps those relationships may change with the age. The bone areas examined by densitometry also differed from those in our study, including the lumbar spine and the femur. Kurt et al. [23] conducted their study only on postmenopausal Caucasian women in Turkey, showing that the presence of the rs731236 (TaqI) polymorphism had not affected the BMD of either the femoral neck or the TH. The reason for the observed differences may be the fact that the Turkish population is not homogeneous for numerous admixtures with other populations [24,25]. A study by González-Mercado et al. [26] conducted on a group of 320 Mexican postmenopausal women showed no differences in bone density in the context of BsmI, ApaI or TaqI polymorphisms. Also, in that case, the differences can be explained in a similar way. Interestingly enough, the authors of another paper [17] describing a group of 300 healthy Saudi women showed that the patients who carried the GG allele for the *VDR* gene rs731236 polymorphism demonstrated a higher risk of osteopenia. This does not coincide with our observations, in which the GG homozygotes revealed higher bone density levels than the other groups. The effect of TaqI on the bone status was not confirmed in a study by Gasperini et al. on a Caucasian group [15], either. It is worth noting, however, that the study group was heterogeneous in terms of sex and age, although it also belonged to the Caucasian population, like the patients in our study. Moreover, the subcohort on which the authors performed the VDR expression analysis was limited to 50 subjects only. Another study conducted on 602 postmenopausal Caucasian women [27] proved, on the other hand, that GG rs731236 homozygotes were associated with an increased risk of osteoporosis. In contrast, AA homozygotes were associated with an increased risk of vitamin D3 deficiency.

Another interesting observation from our study is that the participants under vitamin D and calcium supplementation demonstrated lower BMD values. Marozik et al. [27] also addressed this topic, proving that low bone density was correlated with higher serum vitamin D values. Moreover, the authors demonstrated the same observation for all the polymorphisms analyzed in the study (including rs7975232, rs1544410 and rs731236), suggesting the existence of different mechanisms of vitamin D effects on bone cells and in blood serum. The authors think that the analysis of polymorphisms for rs7975232 (ApaI), rs1544410 (BsmI) and rs731236 (TaqI) may help identify patients with an increased risk of postmenopausal osteoporosis at normal vitamin D levels. This would mean that the one-size-fits-all principle is misguided for vitamin D supplementation, and it would be more appropriate to evaluate *VDR* polymorphism in combination with environmental factors such as exposure to sunlight, physical activity, diet, and concomitant diseases. In addition, the effect of vitamin D may also depend on other genes appearing in the synthesis, activation, and action pathways, as well as epigenetic factors.

The relationship shown in our study that the subjects taking vitamin D supplementation and calcium preparations had lower bone mineral density can be explained by the fact that some of them used that supplementation precisely for therapeutic purposes rather than for preventive ones. On the other hand, serum vitamin D levels alone were correlated positively with BMD and T-score values for both TH and femoral neck scores. This means that the serum vitamin D levels were correlated with bone parameter values, as expected, although the correlation coefficient was not high (r = 0.13–0.16).

Following the American Society for Bone and Mineral Research guidelines, normal serum vitamin D levels are above 25 ng/mL [28]. In our study, none of the participants reached this cutoff point, although 13% and 14% of the participants received vitamin D and calcium supplementation, respectively. Importantly, all the study components (densitometry and blood sampling) were conducted in a short period of time (4 weeks) in May. Thus, the exposure to sunlight was similar for all the participants (the same latitude), modified only by individual factors.

Our study also showed a positive correlation of both BMI and WC with BMD values. That observation was consistent with the previous reports [29,30,31,32]. Nevertheless, the results of the GLOW study published in 2011 disproved the theory regarding the protective role of obesity in preventing bone fractures. The pathochemistry of fractures in obese individuals has been discussed [33]. The study followed more than 40,000 postmenopausal women for two years, 23% of whom met the criteria for obesity. The study found that obesity was associated with an increased risk of fractures, especially in the ankle and the upper leg.

A study on a cohort of nearly 50,000 Swedish men and women (mean age of 70 years) at a median follow-up of 8.7 years proved that for optimal protection against bone fractures, a baseline low BMI and abdominal obesity should be avoided, but also, interestingly enough, a significant reduction in BMI (so-called BMI losers—defined in the study if there was a reduction in BMI values of 0.5 kg/m^2^ over an average of 12 years) [34]. The data on the relationship between abdominal obesity and BMD values are also inconsistent [35,36,37]. In conclusion, the risk of bone fractures, despite higher BMD and T-score values, in people with abdominal obesity and metabolic syndrome remains high. In order to better objectify the qualitative studies on bone parameters, some authors recommend supplementing the diagnosis with ultrasound (QUS) or computed tomography (qCT) of bones [38,39,40]. The effectiveness of vitamin D supplementation in obese individuals also seems important [41]. The accumulation of vitamin D in adipose tissue stores and secondary hyperparathyroidism are considered responsible for the occurrence of vitamin D deficiency in obese people (up to 43% of patients with morbid obesity suffer from secondary hyperparathyroidism) [42].

Last but not least and worth mentioning is the large meta-analysis of *VDR* polymorphisms and postmenopausal women conducted by Lijuan Fu et al. [43]. Out of 507 papers retrieved from Medline, Embase, Wanfang, VIP and CNKI, a total of 67 studies were taken into account, including case-controlled or cohort designed studies and those projects that provided genotypic prevalence rates of VDR polymorphisms in cases with postmenopausal osteoporosis and population-based controls. As a result, the authors concluded that the polymorphisms of rs7975232, rs1544410 and rs731236 were significantly associated with postmenopausal osteoporosis risk in the Caucasian population, whereas the rs10735810 polymorphism was related to the risk of postmenopausal osteoporosis in Asians. The advantage of our study was the large study group, being homogeneous in terms of ethnicity, sex and age (which excludes the influence of hormonal factors), with the additional information concerning modifiable and non-modifiable factors affecting bone mineral density. The assessment of vitamin D concentration and densitometric examination were performed in all the patients in May, and the entire period of material collection lasted up to four weeks. That allowed us to assess the vitamin D concentration in the same period and at the same geographical location. A study limitation was the inability to verify the actual number of fractures in the available medical records or imaging study results—the data on past fractures are based on the testimony of the study participants, so there was a risk of skipping clinically silent fractures in the analysis.

## 5. Conclusions

Our results may contribute to a better understanding of the genetic basis of osteoporosis development. The unique character of our study results from the material: a representative 10% cohort of randomly selected postmenopausal women residing in the Upper Silesia region. The population of women studied was ethnically homogeneous. All the patients belonged to a homogeneous Caucasian population. All that made it possible to obtain unequivocal results from genetic studies.

The results of our study demonstrated that, out of the three studied polymorphisms, only rs731236 (TaqI) genotypes differed among DXA results. The AG heterozygotes were characterized by the lowest median bone mineral density and T-scores. The patients with the AG genotype for *VDR* rs731236 (TaqI) appeared to be more prone to osteoporosis. Parameters such as the age, vitamin D3 concentration, BMI and WC, as well as comorbidities, did not differ among the patients with particular genotypes of the rs731236 (TaqI) polymorphism of the *VDR* gene.

Genetic profiling may serve as a valuable tool for osteoporosis prevention in the future, allowing for the identification of women in need of early treatment, thereby enhancing the level of care in this patient group. Consideration should also be given to expanding research to identify genetic markers associated with the osteoporosis risk in the context of comorbidities.

## Figures and Tables

**Figure 1 biomedicines-12-00917-f001:**
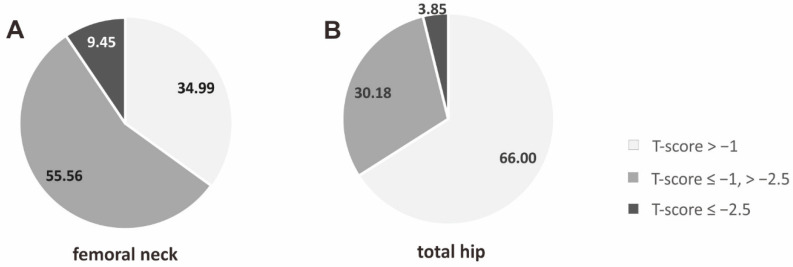
The frequency (%) of individual ranges of T-scores for bone mineral density of the femoral neck (**A**) and total hip (**B**). Legend: BMD, bone mineral density.

**Figure 2 biomedicines-12-00917-f002:**
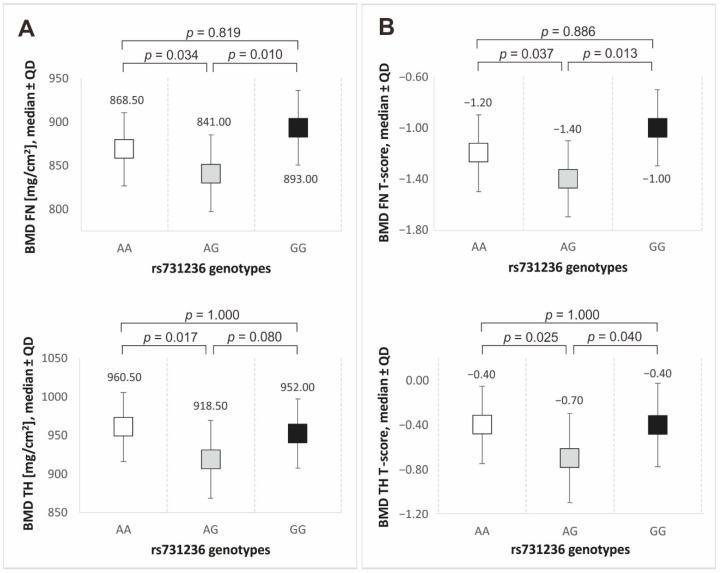
Median values of bone mineral density (**A**) and T-score values (**B**) for rs731236 (TaqI) polymorphism genotypes. Legend: BMD, bone mineral density; FN, femoral neck; TH, total hip; QD, quartile deviation.

**Table 1 biomedicines-12-00917-t001:** Demographic, clinical, and biochemical characteristics of the study group.

Characteristics			
General	Number of subjects, *N* (%)	611	(100.00)
	Age [years], median ± QD	65.82	6.29
	BMI [kg/m^2^], median ± QD	30.84	3.96
	Overweight/obesity [BMI ≥ 25], *n* (%)	533	(87.23)
	Obesity [BMI ≥ 30], *n* (%)	335	(54.83)
	Waist circumference [cm], median ± QD	94.00	8.50
	Abdominal obesity [WC ≥ 88 cm], *n* (%)	392	(64.16)
	Cigarette smokers, *n* (%)	70	(11.46)
	Alcohol consumption [≥3 units/day], *n* (%)	4	(0.65)
	Vitamin D3 [ng/mL], median ± QD	7.49	2.76
BMD parameters	BMD FN [mg/cm^2^], median ± QD	855.00	87.00
	BMD FN T-score, median ± QD	−1.30	0.60
	BMD TH [mg/cm^2^], median ± QD	942.00	92.50
	BMD TH T-score, median ± QD	−0.50	0.75
Comorbidities	Diabetes mellitus type 1, *n* (%)	22	(3.60)
	Diabetes mellitus type 2, *n* (%)	95	(15.55)
	Glucocorticosteroid therapy, *n* (%)	28	(4.58)
	Rheumatoid arthritis, *n* (%)	40	(6.55)
	Thyroid gland diseases, *n* (%)	6	(0.98)
	Chronic kidney disease, *n* (%)	7	(1.15)

Legend: BMD, bone mineral density; BMI, body mass index; FN, femoral neck; TH total hip; QD, quartile deviation; WC, waist circumference.

**Table 2 biomedicines-12-00917-t002:** Genotype and allele frequencies of the analyzed *VDR* gene polymorphisms.

SNP	Position	Genotypes	*n* (%)	Alleles	*n* (%)	HWE *p* Value
rs731236 (TaqI)	chr12:47844974	AA	249 (40.75)	A	773 (63.26)	0.750
		AG	275 (45.01)	G	449 (36.74)	
		GG	87 (14.24)			
		AA + AG	524 (85.76)			
		AG + GG	362 (59.25)			
rs7975232 (ApaI)	chr12:47845054	AA	154 (25.20)	A	613 (50.16)	0.987
		AC	305 (49.92)	C	609 (49.84)	
		CC	152 (24.88)			
		AA + AC	459 (75.12)			
		AC + CC	457 (74.80)			
rs1544410 (BsmI)	chr12:47846052	CC	244 (39.93)	C	763 (62.44)	0.686
		CT	275 (45.01)	T	459 (37.56)	
		TT	92 (15.06)			
		CC + CT	519 (84.94)			
		CT + TT	367 (60.07)			

Legend: HWE, Hardy–Weinberg equilibrium; SNP, single nucleotide polymorphism.

**Table 3 biomedicines-12-00917-t003:** Bone mineral density (BMD) values for rs731236 (TaqI) *VDR* gene polymorphism variants (recessive/dominant model).

SNP	Parameter	Median	±QD	Median	±QD	*p* Mann–Whitney U Test
rs731236		GG		AA/AG		
	BMD FN [mg/cm^2^], median ± QD	893.00	85.50	850.00	85.25	0.028
	BMD FN T-score, median ± QD	−1.00	0.60	−1.30	0.63	0.033

Legend: BMD, bone mineral density; FN, femoral neck; SNP, single nucleotide polymorphism; QD, quartile deviation.

**Table 4 biomedicines-12-00917-t004:** Spearman correlation coefficient (r_s_) values for bone mineral density correlates.

Correlates	BMD FN [mg/cm^2^]	BMD FN T-Score	BMD TH [mg/cm^2^]	BMD TH T-Score	Age [Years]	Vit. D3 [ng/mL]	BMI [kg/m^2^]	WC [cm]
BMD FN [mg/cm^2^]	-	0.98	0.86	0.85	−0.37	0.16	0.28	0.26
BMD FN T-score	0.98	-	0.86	0.87	−0.36	0.15	0.28	0.26
BMD TH [mg/cm^2^]	0.86	0.86	-	0.98	−0.33	0.13	0.45	0.41
BMD TH T-score	0.85	0.87	0.98	-	−0.32	0.14	0.44	0.39
Age [years]	−0.37	−0.36	−0.33	−0.32	-	−0.25	0.12	0.13
Vit. D3 [ng/mL]	0.16	0.15	0.13	0.14	−0.25	-	NS	NS
BMI [kg/m^2^]	0.28	0.28	0.45	0.44	0.12	NS	-	0.87
WC [cm]	0.26	0.26	0.41	0.39	0.13	NS	0.87	-

Legend: BMD, bone mineral density; BMI, body mass index; FN, femoral neck; NS, not statistically significant; TH total hip; QD, quartile deviation; WC, waist circumference; Vit., vitamin.

**Table 5 biomedicines-12-00917-t005:** Potential modifiers of bone mineral density (BMD) parameters.

Modifier		BMD FN [mg/cm^2^]		*p* Mann–Whitney U Test	BMD TH [mg/cm^2^]		*p* Mann–Whitney U Test
		Median	QD		Median	QD	
Obesity [BMI ≥ 30]	Yes	884.50	41.63	0.000	988.00	49.63	0.000
	No	825.00	78.00		896.00	88.50	
Abdominal obesity [WC ≥ 88 cm]	Yes	873.00	82.50	0.000	969.50	94.00	0.000
	No	812.00	79.00		883.00	96.00	
DM type 1	Yes	887.00	87.00	0.215	1016.50	143.50	0.008
	No	855.00	85.50		941.00	91.50	
DM type 2	Yes	880.00	88.00	0.117	984.50	86.00	0.000
	No	853.00	84.50		928.00	94.50	
RA	Yes	812.00	63.00	0.098	875.50	77.50	0.011
	No	858.00	87.00		946.00	93.50	
Calcium supplementation	Yes	837.00	90.00	0.034	883.00	103.50	0.000
	No	859.50	84.75		949.50	91.75	
Vitamin D3 supplementation	Yes	815.00	83.00	0.012	876.00	94.50	0.000
	No	861.00	85.00		952.00	91.50	
Anti-osteoporotic therapy	Yes	832.50	85.50	0.017	876.50	105.00	0.000
	No	859.00	84.50		951.00	91.00	
History of fractures > 40 year	Yes	845.00	84.75	0.029	927.50	92.00	0.017
	No	859.00	84.00		950.00	98.50	

Legend: BMD, bone mineral density; BMI, body mass index; DM, diabetes mellitus; FN, femoral neck; RA, rheumatoid arthritis; TH total hip; QD, quartile deviation, WC, waist circumference.

**Table 6 biomedicines-12-00917-t006:** Distribution of rs731236 (TaqI) polymorphism genotypes depending on the presence or absence of factors modifying bone mineral density.

Modifier		AA		AG		GG		*p* χ^2^ Test	
		*n*	%	*n*	%	*n*	%	Additive Model	Recessive/Dominant
Obesity [BMI ≥ 30]	Yes	129	38.51	158	47.16	48	14.33	0.430	0.944
	No	120	43.48	117	42.39	39	14.13		
Abdominal obesity [WC ≥ 88 cm]	Yes	157	40.05	177	45.15	58	14.80	0.932	0.754
	No	72	41.38	78	44.83	24	13.79		
DM type 1	Yes	9	40.91	12	54.55	1	4.55	0.374	0.185
	No	240	40.75	263	44.65	86	14.60		
DM type 2	Yes	37	38.95	45	47.37	13	13.68	0.881	0.866
	No	212	41.09	230	44.57	74	14.34		
RA	Yes	17	42.50	17	42.50	6	15.00	0.947	0.887
	No	232	40.63	258	45.18	81	14.19		
Calcium supplementation	Yes	39	43.82	34	38.20	16	17.98	0.311	0.275
	No	210	40.23	241	46.17	71	13.60		
Vitamin D3 supplementation	Yes	34	43.04	33	41.77	12	15.19	0.825	0.795
	No	215	40.41	242	45.49	75	14.10		
Anti-osteoporotic therapy	Yes	42	45.65	34	36.96	16	17.39	0.228	0.348
	No	207	39.88	241	46.44	71	13.68		
History of fractures > 40 year	Yes	62	36.47	78	45.88	30	17.65	0.221	0.134
	No	187	42.40	197	44.67	57	12.93		

Legend: BMI, body mass index; DM, diabetes mellitus; RA, rheumatoid arthritis; WC, waist circumference.

## Data Availability

Data are contained within the article.

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
