# Peer review of "Associations between the VDR Gene rs731236 (TaqI) Polymorphism and Bone Mineral Density in Postmenopausal Women from the RAC-OST-POL"

_biomedicines, 2024, doi:10.3390/biomedicines12040917_

Round 1
Reviewer 1 Report
Comments and Suggestions for Authors
The aim of the current study was to comprehensively analyze the polymorphic variants of the VDR gene rs 731236 (TaqI), rs1544410 (BsmI) and rs7975232 (ApaI) in menopausal women with the search for genetic markers to identify patients at a higher risk of developing osteoporosis in the context of their comorbidities.. The effects of the VDR gene polymorphisms on the clinical phenotype of the women studied, including BMD, were analyzed in the context of comorbidities, history data and other clinical factors.
The introduction is well written , with adequate bibliographic references . The objective of the study is clearly established. The methodology is widely described, which would allow the study to be carried out by another research group. The clinical methodology should be more fully described. There are some aspects measured but it is not indicated how they are measured. It would be interesting to comment on whether differences were observed between those who carried out the study and those who decided not to participate. A multivariate study should be carried out where the significant factors are included The results are clear expressed and easy to understand The discussion is adapted to the results obtained. The authors should express the limitations and strengths of the study
Author Response
Dear Reviewer
Thank you for your detailed comments on our work and your time in preparing the review.
Taking into account your suggestion, we have expanded the Conclusions, all the changes made in the manuscript are marked in red font.
We consider that the description of clinical methodology eg. description of group studied is adequate. Please note that we mentioned ref No 22 where more details were available. The manuscript is rather long and additional details would make it even longer. The most important points on population studied eg. its epidemiological nature, size, inclusion criteria and variables gathered are clearly indicated. We consider that in a study with scope on genetic problems such description is sufficient.
We appreciate to suggestion to compare the results in participants and other subjects. However, it is not possible because in the latter subgroups the data lacking.
The results of BMD distribution between rs731236 polymorphism genotypes come from univariate analysis. To ascertain whether other factors differentiating BMD (Table 5) influence our observation, we examined their distribution across genotypes (Table 6). This analysis did not reveal statistically significant differences. Our approach aimed to avoid categorizing quantitative variables (four BMD parameters), necessary for logistic regression analysis. That would entail an entirely different analysis with lower precision compared to the analysis of quantitative variables.
Best regards,
Sylwia GórczyÅ„ska-Kosiorz and the Co-Authors
Reviewer 2 Report
Comments and Suggestions for Authors
The researchers studied a Polish population of postmenopausal women and found that the rs731236 polymorphism of the vitamin D receptor in AG heterozyotes had the lowest bone mineral density compared to two other polymorphisms.
Please see my comments in the attached pdf. Overall, the study is acceptable, but I recommend expert English language editing.

Comments on the Quality of English LanguageAuthor Response
Dear Reviewer
Thank you for submitting your review and valuable comments and your time in preparing the review.
All suggestions regarding the English language in the PDF have been taken into account and corrected. We have also performed professional editing in English. All changes made to the manuscript have been marked in red font.
Best regards,
Sylwia GórczyÅ„ska-Kosiorz and the Co-Authors
Reviewer 3 Report
Comments and Suggestions for Authors
The manuscript titled "The rs731236 (TaqI) polymorphism of the VDR gene affects bone mineral density in postmenopausal women from epidemiologic population of RAC-OST-POL Study: a retrospective cohort study." showcases commendable research efforts by the authors. This study comprehensively analyzes polymorphic variants of the vitamin D receptor (VDR) gene, specifically rs731236 (TaqI), rs1544410 (BsmI), and rs7975232 (ApaI), in menopausal women. The aim is to identify genetic markers associated with osteoporosis risk in the context of comorbidities.
The manuscript achieves a commendable level of clarity and readability, with precise materials and methods, concise discussion, and up-to-date references. The inclusion of high-quality figures and tables enhances comprehensibility.
However, there are key aspects requiring clarification before publication. The title, while descriptive, is lengthy and includes abbreviations (e.g., VDR, RAC-OST-POL) that may not be immediately clear to all readers. Additionally, the conclusion section is concise and would benefit from elaboration. The authors should underscore unique findings and consider recommendations for managing patients with rs731236 (TaqI) genotypes.
In conclusion, while promising, addressing these points is crucial to enhance the manuscript's impact and suitability for publication.
Author Response
Dear Reviewer
Thank you for the detailed review and valuable feedback on our manuscript. We are pleased that you found our manuscript to be clear and concise.
We understand the comments regarding the current title. We propose a new title:
Associations between the VDR gene rs731236 (TaqI) polymorphism and bone mineral density in the postmenopausal women from the RAC-OST-POL.
However, we believe that the abbreviation of the study: RAC-OST-POL should remain in the title because it uniquely identifies the RAC-OST-POL epidemiological study that we started earlier (ref No. 22) and represents its continuation .
Taking into account your suggestion, we have expanded the Conclusions, stated why we believe the results are unique results we proposed the use of rs731236 (TaqI) genotypes in genetic profiling which may affect personalized treatment and better medical care for postmenopausal patients.
All changes made in the manuscript are indicated in red font.
Best regards,
Sylwia GórczyÅ„ska-Kosiorz and the Co-Authors
Round 2
Reviewer 1 Report
Comments and Suggestions for Authors
The authors have made an effort to improve the quality of the article and it provides interesting information for the understanding of the metabolic bone disease. It may be accepted in its current format.
Reviewer 3 Report
Comments and Suggestions for Authors
Well done.